# Immunomodulation of Pluripotent Stem Cell-Derived Mesenchymal Stem Cells in Rotator Cuff Tears Model

**DOI:** 10.3390/biomedicines10071549

**Published:** 2022-06-29

**Authors:** Jieun Baek, Bokyeong Ryu, Jin Kim, Seul-Gi Lee, Min-Seok Oh, Ki-Sung Hong, Eun-Young Kim, C-Yoon Kim, Hyung-Min Chung

**Affiliations:** 1Departmentof Stem Cell Biology, School of Medicine, Konkuk University, Seoul 05029, Korea; jieuns2322@konkuk.ac.kr (J.B.); maxwisdom@konkuk.ac.kr (S.-G.L.); oms860531@konkuk.ac.kr (M.-S.O.); 2College of Veterinary Medicine, Konkuk University, Seoul 05029, Korea; hobitmilk@snu.ac.kr (B.R.); jk208019@snu.ac.kr (J.K.); 3Department of Laboratory Animal Medicine, College of Veterinary Medicine, Seoul National University, Seoul 08826, Korea; 4Advanced Analysis Center, Korea Institute of Science and Technology (KIST), Seoul 02792, Korea; 5Mireacellbio Co., Ltd., Seoul 04795, Korea; kshong@miraecellbio.com (K.-S.H.); jlokey@miraecellbio.com (E.-Y.K.)

**Keywords:** rotator cuff tears (RCT), pluripotent stem cell (PSC), mesenchymal stem cell (MSC), inflammation, treatment

## Abstract

Background: Rotator cuff tears (RCTs) induce chronic muscle weakness and shoulder pain. Treatment of RCT using surgery or drugs causes lipid infiltration and fibrosis, which hampers tissue regeneration and complete recovery. The pluripotent stem cell-derived multipotent mesenchymal stem cells (M-MSCs) represent potential candidate next-generation therapies for RCT. Methods: The difference between M-MSCs and adult-MSCs was compared and analyzed using next-generation sequencing (NGS). In addition, using a rat model of RCT, the muscle recovery ability of M-MSCs and adult-MSCs was evaluated by conducting a histological analysis and monitoring the cytokine expression level. Results: Using NGS, it was confirmed that M-MSC was suitable for transplantation because of its excellent ability to regulate inflammation that promotes tissue repair and reduced apoptosis and rejection during transplantation. In addition, while M-MSCs persisted for up to 8 weeks in vivo, they significantly reduced inflammation and adipogenesis-related cytokine levels in rat muscle. Significant differences were also confirmed in histopathological remission. Conclusions: M-MSCs remain in the body longer to modulate immune responses in RCTs and have a greater potential to improve muscle recovery by alleviating acute inflammatory responses. This indicates that M-MSCs could be used in potential next-generation RCT therapies.

## 1. Introduction

Rotator cuff tears (RCTs) are commonly associated with fibrosis, muscle atrophy, and inter- and intra-muscular fat infiltration, which are commonly referred to as “fatty degeneration” [1]. They are one of the most common shoulder injuries, with an age-related increase in incidence, in addition to shoulder pain and joint dysfunction [2]. Currently, many surgeries are performed for the treatment of RCT. Various surgical treatments are available, such as autografting of the long biceps tendon (LBT) and vasculature-preserved subacromial bursa, and the use of various bioinductive scaffolds. These surgical techniques can alleviate the lesion and have a positive effect on the patient’s prognosis. Nevertheless, some patients suffer from incomplete healing. Fatty degeneration cannot be reversed with surgery, and 90% of muscle fibers are damaged due to shortening of the muscle [1]. The rate of re-tear after surgery was reported to be 7–57% [3]. It also causes additional lipid infiltration and fibrosis, which limit tissue regeneration and prevent complete recovery [4]. Persistent pain and disability also occur [5]. Under this situation, we designed our research to find a wider range of treatments, which could support the surgery and ultimately improve the prognosis.

Stem cell-based therapy can be used to treat fatty degeneration and inflammation. Mesenchymal stem cells (MSCs) are a population of multipotent adult stem cells found in multiple compartments of the body. It has been reported that MSC has an inhibitory effect on the proliferation, differentiation and activation of immune cells and can be usefully applied to the treatment of inflammation-related diseases [6]. MSCs can secrete growth factors to enhance tissue regeneration [7]. However, for adult MSCs, the collection process is invasive, inefficient, and tends to be less effective depending on donor and origin. It also has limitations due to its limited proliferative capacity during ex vivo expansion. On the other hand, human pluripotent stem cell (hPSC)-derived MSCs can be cultured indefinitely, and exceed the limits of clinical application by culturing differentiated MSCs alone [8]. In addition, M-MSCs have a high engraftment rate and low immunogenicity compared to other adult-MSCs, offering high potential for use as a cell therapy agent [9]. Stem cell-based cell therapy always raises safety concerns due to its ability to form teratomas and other tumors, potential immune responses and risk of differentiation into unwanted cell types. However, M-MSCs showed no evidence of adverse events such as abnormal growth, tumor formation, or immune-mediated transplant rejection for 12 months in IC/BPS [10]. In addition, cell safety has been verified by longitudinally monitoring the biodistribution and phenotypic characteristics of M-MSCs injected through confocal microscopy and micro-endoscopy for 6 months in live animals after transplantation [11].

In this study, we hypothesized that M-MSCs would be good candidates for stem cell-based therapies, and that they would affect inflammation relief and muscle recovery in the RCT model. Therefore, we evaluated the transplant suitability of adult-MSCs and M-MSCs using a principal component analysis (PCA) and gene ontology (GO) analysis, and evaluated their ability to suppress immune rejection through mixed lymphocyte response. In addition, we investigated the potential of M-MSCs as a next-generation stem cell-based therapeutic agent in a rotator cuff tear model of M-MSCs by conducting an histological and immunohistochemical analysis and gene expression analysis by creating an RCT model.

## 2. Materials and Methods

### 2.1. Isolation and Culture of M-MSCs

The M-MSCs were obtained from embryoid bodies (EBs) of hPSCs as previously described [12,13]. Briefly, SNUhES cells between passage 40–60 were cultured in DMEM/F-12 Medium supplemented with 20% knockout serum replacement, 1 mM glutamine, 0.1 mM β-mercaptoethanol, 0.1 mM nonessential amino acids, and 4 ng/mL human recombinant bFGF, all of which were purchased from Invitrogen Corporation (Carlsbad, CA, USA) (hPSC culture media). All of the cells were cultured at 37 °C in a humidified atmosphere with 5% CO_2_.

For EB formation, the hPSC colonies were harvested using dispase treatment (1 mg/mL in a serum-containing medium; Roche, Basel, Switzerland) and were grown in suspension culture for 2 days with the same hPSC culture medium except for bFGF. To isolate MSC-like cells, porous membrane transwell inserts with 8 μm pores were used. The upper compartment of the inserts was coated with 0.1% gelatin, and the EBs were attached in EBMTM-2 Basal Medium supplemented with EGMTM-2 MV Microvascular Endothelial Cell Growth Medium Supplements SingleQuotsTM (Lonza, Basel, Switzerland) (EGM2 MV media) for 5 days. The cells that migrated to the lower compartment of the inserts formed colonies, which were gently scraped and subcultured onto a new 100 mm dish in the same EGM2 MV media, and were called M-MSCs.

The M-MSCs were cultured in EGM2 MV media and were expanded for fewer than 10 passages to ensure that multipotency was preserved. The medium was changed every two days. Each cell was harvested when it reached 95% confluency.

### 2.2. RNA-Sequencing for M-MSCs

RNA-sequencing (R-seq) was conducted to compare M-MSCs to bone marrow-derived mesenchymal stem cells (BM-MSCs) and adipose-derived mesenchymal stem cells (AD-MSCs). Firstly, total RNA was extracted from the M-MSCs using GeneAll^®^ Ribospin™ (GeneAll, Seoul, Korea) according to the manufacturer’s protocol and all RNA samples were then determined as high and comparable quality. Libraries were generated according to standard procedures using a TruSeq Stranded mRNA LT Sample Prep Kit (Illumina Inc., San Diego, CA, USA) according to a TruSeq Stranded mRNA Sample Preparation Guide, Part #15031047 Rev. E (Illumina Inc.). The libraries were sequenced on a NovaSeq 6000 S4 (Illumina Inc.) according to a NovaSeq 6000 System User Guide Document #1000000019358 v02 (Illumina Inc.) in a sequencer NovaSeq (Illumina Inc.). The set of RNA-seq data for M-MSCs was derived from five biological samples.

The trimmed reads scoring values of over 98% of the average Q20 and over 95% of the average Q30 in Phred quality score using BBDuk (part of BBtools) were kept. The remaining reads were mapped to the reference genome sequence (University of California Santa Cruz (UCSC) hg19, annotation RefSeq_2017_06_12) using Bowtie2 [14]. Calculation reads were counted using Bedtools (https://bedtools.readthedocs.io/en/latest/ accessed on 7 February 2022). Read mapping and expression quantification were performed separately for each sample.

Additionally, from National Center for Biotechnology Information Gene Expression Omnibus (NCBI GEO), transcriptomic data of BM-MSCs (GEO numbers: GSM5068578 and GSM5068579 in GSE166327; GSM2934991 and GSM2934992 in GSE109181) and AD-MSCs (GEO numbers: GSM4873400 in GSE160439; GSM4820187 and GSM4820190 in GSE159137) were obtained. The gene expression levels then were quantile normalized using R 4.0.3 (The R Foundation for Statistical Computing c/o Institute for Statistics and Mathematics, Vienna, Austria).

### 2.3. Transcriptomic Analysis for M-MSCs Compared to Adult MSCs

Pearson’s product–moment correlation coefficient rho (*r*) and significance levels were calculated using the function rcorr() and the function chart. Correlation() found in the R statistical programming language and the correlogram was plotted using the R packages. The strength of associations were categorized as follows: 0.90 to 1.00, very highly correlated; 0.70 to 0.90, highly correlated; 0.50 to 0.70, moderately correlated; 0.30 to 0.50, lowly correlated; and 0.00 to 0.30, negligibly correlated [15]. Additionally, principal component analysis (PCA) was performed to visualize and quantify multi-dimensional variation between M-MSCs, BM-MSCs, and AD-MSCs. Principal components were calculated using the function prcomp() found in the R statistical programming language and plotted using the function autoplot() (pam) in R packages.

Upregulated differentially expressed genes (DEGs) in the M-MSCs were selected based on fold changes (FCs) 4 times higher than BM-MSCs or AD-MSCs while downregulated DEGs based on FCs were 0.25 times lower. For the selected DEGs, a Gene Ontology (GO) network and Enrichment Pathway analysis were performed using ClueGO 2.5.7 and CluePedia 1.5.5 on Cytoscape 3.7.2 (Cytoscape Consortium; San Diego, CA, USA) on Java script 1.8.0_162 (Oracle Corporation; Santa Clara, CA, USA) [16]. ClueGo analyses incorporated GOs for GO_BiologicalProcess-EBI-UniProt-GOA-ACAP-ARAP_08.05.2020_00h00: 17972 (The European Bioinformatics Institute; EMBL-EBI, Cambridgeshire, UK). The pathway’s restriction was set to a *p* < 0.05, and a GO tree interval with a minimal from 4 to maximal of 6 was used to specify GO terms. The minimal number of genes in a GO was set to 3 while the connectivity score (Kappa Score) was set to 0.4.

Additional analyses for specific GOs were conducted with the GO term GO: 0043069 ‘negative regulation of programmed cell death’ (EMBL-EBI) and GO term WP2328 ‘Allograft Rejection (Homo sapiens)’ (https://wikipathways.org accessed on 10 February 2022) [17]. The genes in M-MSCs were transformed into log2-FCs against BM-MSCs or AD-MSCs for the specific GOs. Quality Threshold Clustering (QTC) was then applied to the genes with Pearson’s correlation matrix, a cluster diameter of 0.5, and at least 5 genes per cluster using the TIGR MultiExperiment Viewer (MeV) 4.9.0 (https://sourceforge.net/projects/mev-tm4/files/mev-tm4 accessed on/ 21 May 2021).

### 2.4. Mixed Lymphocyte Reaction (MLR) for M-MSCs, BM-MSCs, and AD-MSCs

Splenocytes were isolated from mouse spleen by disaggregation into 10 mL RPMI 1640 medium. Erythrocytes were lysed with RBC lysis buffer (Roche, Munich, Germany) and subsequently washed 3 times in RPMI 1640. Stimulator splenocytes (10^7^ cells/mL) were treated with 10 μg/mL mitomycin C at 37 °C for 1 h, followed by 5 extensive washes with FBS-containing RPMI 1640 medium. Responder splenocytes from BALB/c mice and stimulator splenocytes from different strains of mice were resuspended in RPMI 1640 containing 10% FBS, 2 mM glutamine, 100 U/mL penicillin, 100 μg/mL streptomycin, 1 mM sodium pyruvate, 20 mM HEPES, and 50 uM 2-mercaptoethanol. Each responder and stimulator cell population were seeded in triplicate at a concentration of 10^5^ cells/100 μL/well, in 96-well. MSCs were also added to the MLR at the same number to obtain a 300-μL final volume. In mitogen proliferative assays, responder splenocytes were incubated with 5 μg/mL concanavalin A. After 5 days, the proliferation of responder cells was measured using a CCK assay.

### 2.5. Rotator Cuff Tear (RCT) Model of Rats

A total of one hundred and sixty male 7-week-old Sprague Dawley (SD) rats were purchased from ORIENT Bio Inc. (Gyeonggi, Korea). These rats were acclimated for a week and used at the age of 8-weeks-old. During the experiment, all rats were provided with tap water and pellet (Altromin, Lage, Germany) ad libitum. All animals were maintained at a temperature of 22 ± 2 °C with a humidity of 55 ± 5% and a light/dark cycle of 12 h/12 h.

The RCT model was induced according to the previously described method [18] with some modifications. One hundred and sixty rats were randomly divided into 20 groups (M-MSC, BM-MSC, Control, Normal for each time; n = 8) and received unilateral complete tendon detachment under anesthesia with Isoflurane. Briefly, in the right sides shoulder, a lateral skin incision was made to expose the supraspinatus and infraspinatus muscles at their insertions into the greater tuberosity of the humerus. The RCT model was created by incising 50% of the supraspinatus and infraspinatus muscles to reveal the tendons and cutting the supraspinatus and infraspinatus tendons from the greater tuberosity. After surgery, M-MSCs, BM-MSCs (PCS-500-012, ATCC) and human dermal fibroblasts (hDFs; CCD-986sk, Korean cell line bank) were injected into the supraspinatus and infraspinatus muscles at 3 × 10^6^ cells per animal. The normal group refers to no surgery or cell treatment, and the control group refers to the hDF-treated group as a control for cell treatment. The skin was closed with 6-0 nylon suture, without repairing the tendon. The rats were allowed unrestricted cage activity. At 24 h, 72 h, 7 days, 4 weeks, and 8 weeks after surgery, each rat was euthanized with CO_2_ gas inhalation.

### 2.6. Long-Term In Vivo Cell Tracking Using DiI

M-MSCs, BM-MSCs and hDF cells were adhered to and cultured in a 100 mm dish, and when the density reached 90%, the media was removed and washed with DPBS. Cell Tracker™ CM-DiI dye (Thermo fisher, Waltham, MA, USA) stock was prepared as recommended by the manufacturer. The stock was diluted to 1 μM in DPSB and placed in a cell dish. Incubation was performed at 37 °C for 15 min and then at 4 °C for 15 min. Then, the staining solution was removed, washed twice with DPBS, and the cells were detached using trypsin. The cell suspension was centrifuged, adjusted to 3 × 10^6^ cells, and 200 μL was prepared and injected at the time of surgery. After sampling at 24 h, 72 h, 7 days, 4 weeks, and 8 weeks, paraffin sections were performed, and the labeled cells were monitored using a microscope (Eclipse Ti, Nikon, Tokyo, Japan).

### 2.7. Grip Strength Measurement in Rats

A grip strength meter (Bio-GS3, BioSeb, Vitrolles, France) was used to assess the forelimb grip strength of the rats. Rats were lifted by the tail and induced to grasp a mesh attached to a digital force gauge with its right forelimb. The mesh was 3 × 4.75 square inches with 1-square inch grids. The bars of the grids were 1/8-inch thick. The mesh was oriented horizontally. The animal was gently lifted over the mesh by holding its tail and then lowered toward the mesh while its tail, body, and forelimb were kept in a line and perpendicular to the mesh when the animal reached out to hold onto the mesh. The tension reading of the digital force gauge was defined as the grip strength before the rat released the net. Three consecutive tests were performed on each rat and the mean maximum limb muscle strength value (grams; g) was obtained. It was measured at 24 h, 72 h, 7 days, 4 weeks, and 8 weeks after surgery, and the value that reached the normal measured value was defined as a percentage.

### 2.8. RT-qPCR Analysis to Monitor Physiological Status of In Vivo Muscles

Total RNA was extracted from the supraspinatus and infraspinatus muscles using GeneAll^®^ Ribospin™ (GeneAll, Seoul, Korea). First-strand cDNA was then synthesized using the extracted total RNA as a template with AccuPower^®^ RT PreMix (Bioneer, Deajeon, Korea) according to the manufacturer’s protocols. The resulting cDNA was subjected to qPCR using AccuPower^®^ 2× GreenStar™ qPCR Master Mix (Bioneer, Deajeon, Korea). Primer sequences and detailed experimental conditions for the amplifications of GAPDH, IL-1b, IL-6, TNF-a, Cox2, Ngf, Ptges, Ppar-r, MyoD, Myf5 and cebpa are shown in Table 1. All primers were purchased from Macrogen (Seoul, Korea). Cycle threshold (Ct) values from each sample were normalized to those of GAPDH as an internal control (ΔCt = Ct_target gene_ − Ct_GAPDH_). Relative fold changes of target gene expressions were determined using the comparative 2^−ΔΔCt^ method (ΔΔCt = ΔCt_M-MSC_ − ΔCt_control_) [19].

### 2.9. Antibody-Based Protein Microarray

Whole muscular tissues from the RCT models treated with M-MSCs, BM-MSCs or hDFs were excised at 72 h after surgery for comparison with a normal rat. For the semi-quantitative protein antibody array, proteins (1300 to 1800 μg) were extracted from the muscular tissues. A Rat L2 Antibody Array slide (RayBiotech Inc., Norcross, GA, USA) was dried for 2 h at room temperature and was incubated with 400 μL of blocking solution at room temperature for 30 min. After decanting the blocking buffer from each sub array, 400 μL of the diluted samples was added and samples were incubated for 2 h at room temperature. After decanting the samples, each array was washed three times with 800 μL of 1× wash buffer I at room temperature for 5 min with shaking. The glass chip assembly was placed into the container and a sufficient amount of 1× wash buffer I was added to submerge the entire glass chip for 10 min with shaking twice. The advanced washing step with 1× wash buffer II was repeated. 1× biotin-Conjugated Anti-Cytokine antibodies were prepared and incubated for 2 h at room temperature with gentle shaking, and washed with 150 μL of 1× wash buffer I at room temperature with shaking. A 1× Cy3-Conjugated Streptavidin stock solution was added and incubated for 2 hr at room temperature with gentle shaking, and was washed with 1× wash buffer I for 10 min at room temperature twice. After washing, the slide was rinsed with deionized water using a plastic wash bottle and was centrifuged at 1000 rpm for 3 min to remove water.

The slide scanning was performed using a GenePix 4100A Scanner (Axon Instrument Inc., Foster City, CA, USA). The slides were completely dried before the scanning and scanned within 24-48 h. The slides were scanned at a 10 μm resolution, with optimal laser power, and PMT. After obtaining the scanned image, they were gridded and quantified with GenePix Software (Axon Instrument Inc., Foster City, CA, USA). After analysis, the data about protein information were annotated using UniProt DB and were quantile normalized.

### 2.10. Histopathological Analysis of In Vivo Muscles

To evaluate histopathological changes, the supraspinatus and infraspinatus muscles of each rat were freshly excised and fixed with 10% neutral buffered formalin for 24 h. Tissues were processed using routine tissue techniques and embedded in paraffin in cross-sections for the supraspinatus and longitudinal sections for the infraspinatus. Paraffin-embedded specimens were sliced into 5-μm-thick sections. Sections were then transferred to adhesive microscope slides (Marienfeld, Lauda-Königshofen, Germany). Deparaffinized muscle sections were stained with hematoxylin and eosin (H and E), Masson’s trichrome (MT) and Toluidine blue (TB). Additionally, immunohistochemical staining (IHC) was conducted using an anti-CD68 antibody (Abcam, Cambridge, UK). All stained sections were then examined with a light microscope (Eclipse Ti, Nikon, Tokyo, Japan) to assess histological changes including mast cell infiltration, and CD68-positive macrophage infiltration. Three sections per animal were used for histological examinations.

### 2.11. Statistical Analysis

The transcriptomic data were analyzed to find their Pearson product–moment correlation coefficient rho (r) and their significance levels using the functions rcorr() and chart.Correlation() within R 4.0.3 (The R Foundation for Statistical Computing c/o Institute for Statistics and Mathematics, Vienna, Austria), and the PCA was conducted using the functions prcomp() and autoplot() (pam) within R. The values of the correlation and significance level are displayed for those with a P value of less than 0.01.

After the selection of the upregulated DEGs in the M-MSCs based on FC > 4 and downregulated DEGs based on FC < 0.25, the selected DEGs were analyzed using ClueGO 2.5.7 and CluePedia 1.5.5 on Cytoscape 3.7.2 (Cytoscape Consortium; San Diego, CA, USA) on Java script 1.8.0_162 (Oracle Corporation; Santa Clara, CA, USA) with GO_BiologicalProcess-EBI-UniProt-GOA-ACAP-ARAP_08.05.2020_00h00: 17972 (The European Bioinformatics Institute; EMBL-EBI, Cambridgeshire, UK), a *p* < 0.05, GO trees between 4 and 6, minimal = 3 genes per node, and Kappa Score = 0.4.

Additionally, the transcriptomic data were transformed with log2-FCs and Quality Threshold Clustering (QTC) was then applied using the TIGR MultiExperiment Viewer (MeV) 4.9.0 (TM4 Software Suite, USA) with Pearson correlation matrix, a cluster diameter = 0.5, and minimal = 5 genes per cluster. The data are shown as mean ± standard deviation (SD).

While the data of qPCR are shown as mean ± standard error of the mean (SEM), the data of the MLR, the grip strength, the areas of fibrosis, the number of mast cells, and the number of CD68+ macrophages are shown as mean ± SD. All statistical analyses were performed using GraphPad Prism (v.5, GraphPad software., San Diego, CA, USA). Statistical significance was considered at a *p*-value of less than 0.05.

### 2.12. Ethics Statement

All animal experiments were performed in accordance with relevant guidelines and regulations of the Institutional Animal Care and Use Committee of Konkuk University (IACUC authorization no. KU20008) accredited for laboratory animal care by the Ministry of Food and Drug Safety of South Korea.

## 3. Results

### 3.1. Transcriptomic Characteristics of M-MSCs and Mixed Lymphocyte Reaction

A PCA of several samples of M-MSCs, BM-MSCs, and AD-MSCs was performed to establish the consistency of M-MSCs as cell lines (Appendix A). Samples of BM-MSCs and AD-MSCs tended to cross-cluster with each other due to batch diversity, whereas clusters of M-MSCs were generated only due to cell line consistency. Meanwhile, M-MSCs showed a significant correlation with BM-MSCs overall, as shown in Appendix A (correlation efficient, *r* = 0.42 ± 0.05). However, the correlation between M-MSCs and AD-MSCs was negligible (*r* = 0.25 ± 0.02).

The characteristics of M-MSCs compared with adult MSCs were also confirmed via DEG analysis (Figure 1). As shown in Figure 1A1 and Appendix A, the genes upregulated in M-MSCs compared with BM-MSCs are typically EMBL-EBI GO terms: (1) GO:0050919 ‘negative chemotaxis’, (2) GO:0048762 ‘mesenchymal cell differentiation’, (3) GO:0042060 ‘wound healing’, (4) GO:0061041 ‘regulation of wound healing’, (5) GO:0021882 ‘regulation of transcription from RNA polymerase II promoter’, and (6) GO:1902292 ‘cell cycle DNA replication initiation’. As a result, they were identified as belonging to GOs that negatively regulate inflammation and are involved in tissue repair. Genes downregulated in M-MSCs compared with BM-MSCs included typically GO terms, such as (7) GO:0006342 ‘chromatin silencing’, (8) GO:0016458 ‘gene silencing’, and (9) GO:0097501 ‘stress response to metal ion ‘, which were associated with GOs related to stress response (Figure 1A2 and Appendix A). As shown in Figure 1A3 and Appendix A, the genes upregulated in M-MSCs compared with AD-MSCs were typically GO terms: (10) GO:0050922 ‘negative regulation of chemotaxis’, (11) GO:0060850 ‘regulation of transcription involved in cell fate commitment’, and (12) GO:0008330 ‘protein tyrosine/threonine phosphatase activity’, which were identified as belonging to GOs that negatively regulate inflammation. Finally, as shown in Figure 1A4 and Appendix A, the genes downregulated in M-MSCs compared with AD-MSCs were typically GO terms: (13) GO:0097501 ‘stress response to metal ion’, (14) GO:0006935 ‘chemotaxis’, (15) GO:1902622 ‘regulation of neutrophil migration’, and (16) GO:0045736 ‘negative regulation of cyclin-dependent protein serine/threonine kinase activity’, associated with GOs related to inflammation and stress response. Thus, compared with BM-MSCs and AD-MSCs, GO-related genes that negatively regulate inflammation-related phenomena are upregulated in M-MSCs, and concurrently inflammatory cell-related pathways and stress response GO-related genes are downregulated.

To confirm their application as a cell therapy for transplantation, the characteristics of M-MSCs were analyzed for specific GOs compared with adult MSCs (Figure 1). Regarding GO: 0043069 (EMBL-EBI), a GO related to negative regulation of programmed cell death, M-MSCs showed an overall increase in gene expression compared with BM-MSCs and AD-MSCs. Simultaneously, for WP2328 (WikiPathways), an allograft rejection-related GO, M-MSCs showed an overall decrease compared with adult MSCs. Compared with adult MSCs, M-MSCs show reduced levels of apoptosis and less rejection during transplantation (Appendix A).

Similar to the NGS results, a mixed lymphocyte reaction was performed to determine whether or not M-MSCs had low immunogenicity compared with other adult MSCs. The results showed that all MSCs exhibited limited activity of responder splenocytes caused by stimulator splenocytes. M-MSCs exhibited significantly lower levels of immune rejection than AD-MSCs, and showed immunogenicity comparable to gold standard BM-MSCs (Figure 1C). Based on the results of NGS and MLR, M-MSCs and BM-MSCs with low immunogenicity were set as RCT therapeutic candidates.

### 3.2. Grip Strength Test for Quantification of Rotator Muscular Function Retrieval

We compared the grip strength for each administered cell and time. All values were expressed as percentages of the values reached to the values of the normal group each time. No differences were found between cells at 24 h and 72 h after surgery. Starting from day 7, the MSC group showed faster muscle recovery than the hDF group. The M-MSCs group showed a more significant difference than the BM-MSCs group (Figure 2B). Compared with the normal group, the M-MSCs group showed no difference from day 7, whereas the BM-MSCs group was similar to the normal group only after 8 weeks. (Not shown in the graph.) We confirmed that the M-MSCs group showed much faster and superior muscle recovery than the BM-MSCs.

### 3.3. Morphological Evaluation of Muscles In Vivo Based on H and E and MT Staining

H and E and MT staining were performed for histological analysis. Fat infiltration, histological atrophy, and fibrosis were confirmed via H and E staining. In the supraspinatus muscle, histological atrophy was observed for up to 7 days in the M-MSCs group, whereas no fibrosis or fat infiltration was detected. In the BM-MSCs group, some fat infiltration was observed at 4 weeks, but overall it was less apparent. However, histologic atrophy and muscle fibrosis were observed until week 4. The histological atrophy was not fully reversed at week 8. The hDF group showed serious fat infiltration from day 7, and the fibrosis and muscle atrophy progressed from 72 h until week 8 (Figure 3A1).

The M-MSC group showed atrophy of infraspinatus until week 4, although fibrosis or fat infiltration was confirmed at 7 days, suggesting that the M-MSCs group recovered adequately to approach the normal group from week 4. In the case of the BM-MSCs group, the fibrosis or muscle atrophy recovered after 7 days, similar to that of the M-MSCs group. However, the fat infiltration was severe from day 7 until week 4. In the case of the hDFs group, muscle atrophy, fibrotic muscle, and fat infiltration were observed at up to 8 weeks (Figure 3A2).

Fibrosis was confirmed by MT staining. In the supraspinatus muscle, the M-MSCs and BM-MSCs groups showed a significant difference compared with the normal group, accounting for about 20% of the total area of fibrosis until 72 h. The fibrotic area was reduced and recovered to almost the same value. In contrast, in the hDFs group, a significant range of fibrosis was confirmed at up to 8 weeks (Figure 3B1,B2).

In the infraspinatus muscle, the M-MSCs group was similar to the supraspinatus muscle, and fibrosis was confirmed for up to 72 h, followed by a decline from day 7 and recovery similar to the normal group after 4 weeks. In the case of BM-MSCs, fibrosis increased significantly compared with the M-MSCs group until day 7, and decreased from week 4. In the case of the hDFs group, fibrosis persisted until week 4, and the maximum fibrosis was confirmed on day 7, followed by a decline similar to the normal group at 8 weeks (Figure 3B3,B4). H and E and MT staining revealed reduced muscle atrophy, fibrosis, and fat infiltration in the M-MSCs group in the RCT model.

### 3.4. Inflammatory Cell Counting Based on TB Staining and IHC for CD68

Mast-cell infiltration was confirmed via TB staining. In the case of the M-MSCs group, usually less than 5 mast cells per area were found in the supraspinatus, but there was no significant difference compared with the normal group. In the infraspinatus, fewer than 10 mast cells were found at 24 h. A slightly higher number of mast cells than in the normal group was found, but there was little or no significant difference from the normal group after 72 h. In the case of the BM-MSCs group, mast cells were found in the supraspinatus similar to M-MSCs, without a significant difference from the normal group. Inflammation in the infraspinatus was not significantly different compared with the normal group at up to 72 h, and mast cells were confirmed. In the case of hDFs, a significantly higher number of mast cells were confirmed compared with the normal group at up to 7 days, which affected the early stage after surgery (Figure 4A1–A4).

Next, macrophage infiltration was confirmed by CD68 staining. In the M-MSCs group, macrophages were found with a significant difference from the normal group only at 72 h in both the supraspinatus and infraspinatus. In the case of the hDFs and BM-MSCs groups, macrophages were detected even after 72 h, even on day 7. Almost no macrophages were detected in the supraspinatus and infraspinatus at 24 h in all three groups, and almost no macrophage was detected even after 4 weeks. Both mast cells and macrophages were hardly detected after 4 weeks, but were found at the initial stage before day 7 (Figure 4B1–B4). The M-MSCs group also showed less inflammatory response than other groups.

### 3.5. RT-qPCR Analysis of Genes Associated with Inflammation, Adipogenicity, and Myogenicity

The findings based on behavioral and histological evaluation were quantified via q-PCR of nine representative genes related to inflammation, adipogenesis and muscle synthesis. The mRNA expression of inflammatory genes including TNF-a, IL-6, IL-1b, Cox2, and Ptges in the supraspinatus muscle was significantly decreased in the M-MSCs group until day 7 (Figure 5A1–A5). The expression of the inflammatory genes in the infraspinatus was established at up to 4 weeks and significantly decreased compared with the hDFs and BM-MSC groups, suggesting that M-MSCs rapidly reduced the inflammatory response in RCT (Figure 5B1–B5).

There was no significant difference in the expression of the adipogenic gene, but the expression of Pparg significantly reduced in both the supraspinatus and infraspinatus muscles at 72 h. There was no difference in Myod1 in the supraspinatus, but there was a significant difference in the infraspinatus from week 4 to week 8 (Figure 5A6,A7,B6,B7). The mRNA expression of Myf5, a myogenic regulator, continued to decrease significantly from the early to the late stage, and in the case of Cebpa, a significant decrease was confirmed until the first 7 days (Figure 5A8,A9,B8,B9). These results showed similar tendencies in the antibody array (Appendix A). M-MSCs showed inhibition for inflammation, fat infiltration, and muscle damage.

### 3.6. Long-Term In Vivo Tracking of MSCs Using DiI

Each cell was stained using Cell Tracker™ CM-DiI dye and then injected into the muscle during surgery. Postoperative cell duration and effects on muscle were monitored. It was confirmed that all three groups of cells remained in the muscle until 4 weeks. The number of cells remained in the order of M-MSCs, BM-MSCs, and hDFs. M-MSCs and BM-MSCs were observed intramuscularly for up to 8 weeks (Figure 6), and it was confirmed that they survived significantly longer than hDFs (Appendix A). We confirmed that M-MSCs remained in the muscle for a long time while relieving inflammation and promoting muscle recovery.

## 4. Discussion

M-MSCs are very accessible because of their unlimited source and easy production methods. Even though the cells were derived from hPSCs, M-MSCs present functional MSC features including multi-potency [12]. In addition, establishing the consistency of M-MSC as a cell line facilitates the development of cell therapy products. Based on the principal components extracted from several MSCs, M-MSCs clustered very closely and showed consistency as a cell line, whereas adult MSCs lacked consistency in cell characteristics according to donors. These consistencies of the M-MSCs were proved within several M-MSC lines derived from various hPSC cell lines in a previously published paper, confirming that there is no significant difference between cell lines from deferent donors using flow cytometry and karyotype analysis [12].

Additionally, M-MSCs are relatively superior in regulating the immune response including inflammation. Previously, it was reported that the supernatants of M-MSCs contain more proteins affecting the regulation of inflammation than BM-MSCs, including (1) SMAD protein regulation and (2) negative regulation of cellular responses to oxidative stress. Secretion was confirmed. [13] Based on these results, it was expected that M-MSCs would have the ability to control inflammation through the paracrine effect, so cells were injected into the muscle instead of the completely amputated tendon. To demonstrate these characteristics of the M-MSCs in transcriptomic level, the cells were compared with BM-MSCs and hDFs. BM-MSCs were set as RCT treatment candidates because they confirmed that they had low immunogenicity based on the results of NGS and MLR as the golden standard for MSCs. On the other hand, the hDFs were chosen as a universal negative control as in previously published articles related to cell therapy, and since the M-MSCs we investigated the effect of treatment for were PSC-based cells, we set this as a control because hDF is a basal cell [20].

The immunomodulatory abilities of M-MSCs were demonstrated by the transcriptomic expression levels. Compared with adult MSCs, GOs related to inflammatory inhibition and tissue recovery enhancement including GO:0050919 (negative chemotaxis), GO:0042060 (wound healing), GO:0061041 (regulation of wound healing), and GO:0050922 (negative regulation of chemotaxis) were upregulated in M-MSCs, while GOs related to stress reaction and inflammation including GO:0097501 (stress response to metal ion), GO:0006935 (chemotaxis), and GO:1902622 (regulation of neutrophil migration) were downregulated. Based on these transcriptomic features of M-MSCs, M-MSCs are effective in regulating immune responses such as inflammation as reported previously [13].

GO:0008330 (protein tyrosine/threonine phosphatase activity) was upregulated in M-MSCs compared with AD-MSCs, and included genes related to activities of tyrosine phosphatases and threonine phosphatases. The tyrosine phosphatases regulate the excessive activation of lymphocytes, as shown in the case of the exacerbation of acute and chronic enteritis without tyrosine phosphatases such as PTPN2 and PTPN22 in mice [21]. In addition, threonine phosphatases regulate the inflammatory response, as demonstrated by the induced TLR-triggered immune responses and proinflammatory cytokine production in the knockdown model of threonine phosphatases such as PP1 [22]. Based on the upregulation of inflammation regulatory GO, it is possible to deduce the origin of the mechanism underlying the superior regulatory function of M-MSCs. Additionally, it is possible to determine protein tyrosine/threonine phosphatase activities in additional studies investigating M-MSCs.

M-MSCs not only play an extraordinary role as cell therapeutics, but also survive for a long time based on their intensive immune evasiveness. First, M-MSCs showed an overall decrease in allograft rejection-related GOs compared with adult MSCs such as BM-MSCs and AD-MSCs at the transcriptomic level, and increased tendencies in GOs related to the negative regulation of programmed cell death. In addition, during the application of M-MSCs and adult MSCs in a mixed lymphocyte reaction (MLR) assay in which responder cells reacted with stimulator cells with mismatched histocompatibility antigens [23], M-MSCs acted as a powerful inhibitor similar to BM-MSCs, a gold standard of MSCs, and were significantly stronger than AD-MSCs. Based on the factors inhibiting the MLR, it might be possible to reduce the side effects including graft-versus-host disease (GVHD) [24], as M-MSCs suppress the immune rejection. The usefulness of this powerful immune evasiveness of M-MSCs was indirectly demonstrated in in vivo monitoring for 8 weeks. A significantly higher number of M-MSCs than BM-MSCs survived in the intramuscular space. In a previous study, we found that the expression of genes such as WNT, FOS, and CDK1 was high through single-cell transcriptome analysis of M-MSCs transplanted for IC/BPS treatment [11]. The WNT gene protects tissues from environmental damage and builds a microenvironment favorable for recovery, and the FOS gene promotes adaptation to new cells and microenvironments, which is beneficial for alleviating inflammation. In addition, CDK1 maintains various functions of stem cells and enhances immunomodulatory activity, thereby enhancing therapeutic efficacy. A significant number of M-MSCs remained in the body for a long time and played an excellent therapeutic role based on the high expression of these genes and excellent immune evasion properties. Compared to the existing adult-MSCs that acted for a short time in a ‘hit and run’ method [25,26], M-MSCs remained for a long time and played a therapeutic role, extending the administration period as a cell therapy and lowering the overall treatment cost.

The immunomodulatory ability of M-MSCs was shown to attenuate the acute inflammatory markers of damaged muscle tissue in the RCT model. When comparing the mRNA expression of inflammatory cytokines (TNf-alpha, IL-6, IL-1beta, Cox2, and Ptges), which are mainly detected in RCT, there was a significant difference from 7 days until 4 weeks in both the upper and lower extremities. In the M-MSCs-administered group, a significantly lower or similar level of inflammation was observed compared with the golden standard BM-MSCs. In addition, it was confirmed that the group treated with M-MSCs decreased the most distressing fat infiltration in RCT compared to other groups and that myogenicity, determined by Myf5 and MyoD and acting as an indirect indicator for muscle damage caused by inflammation, decreased. It was confirmed that M-MSCs help in the recovery of the rotator cuff by reducing inflammation, preventing muscle damage, and inhibiting inflammation and fat infiltration.

In the group treated with M-MSCs intramuscularly, the forelimb muscle histology and function was close to normalization. The grip strength was restored by M-MSCs to levels comparable to that of the normal group in 7 days. The BM-MSCs group showed a significant difference in the hDF group from day 7, but the normal level was attained after 8 weeks. Histological atrophy of muscle, fibrosis, and fat infiltration were confirmed via H and E staining. In the M-MSCs group, a sharp decrease was observed after 7 days, and in other cells at up to 4 weeks. The fibrosis observed via MT staining also showed a sharp decrease after day 7 in the M-MSCs group. In the case of mast cells and macrophages, a significant difference was found between the M-MSCs group and other groups. Given that these histological results and the results of forelimb grip strength are similar, M-MSCs facilitate the functional recovery of muscles from an early stage and rapid normalization. The practical clinical application of cell therapy requires the validation of safety and long-term monitoring and tracking at the cellular level. Additionally, we used rats as the RCT model, but since their recovery rate is different from humans, it is necessary to confirm whether the effect is equally potent in humans. Currently, M-MSCs are embryonic stem cell-derived cells and present an ethical problem, so their conversion to induced pluripotent stem cells is required. Therefore, we are working to construct hiPSC-derived M-MSCs. By establishing the protocol, the advantages of other adult-MSCs for culturing cells taken from patients in vitro can be obtained, and it is expected to provide a method for culturing without ethical issues.

## 5. Conclusions

In this study, we confirmed that PSC-derived M-MSCs have superior accessibility and immunomodulatory ability and persist longer in the body compared to the conventional primary culture of adult MSCs. In addition, it was shown that the forelimb function of rats was restored in the RCT model by alleviating the acute inflammatory response by modulating the immune response. Therefore, it was confirmed that M-MSCs can be applied as a next-generation cell therapy in RCT.

## Figures and Tables

**Figure 1 biomedicines-10-01549-f001:**
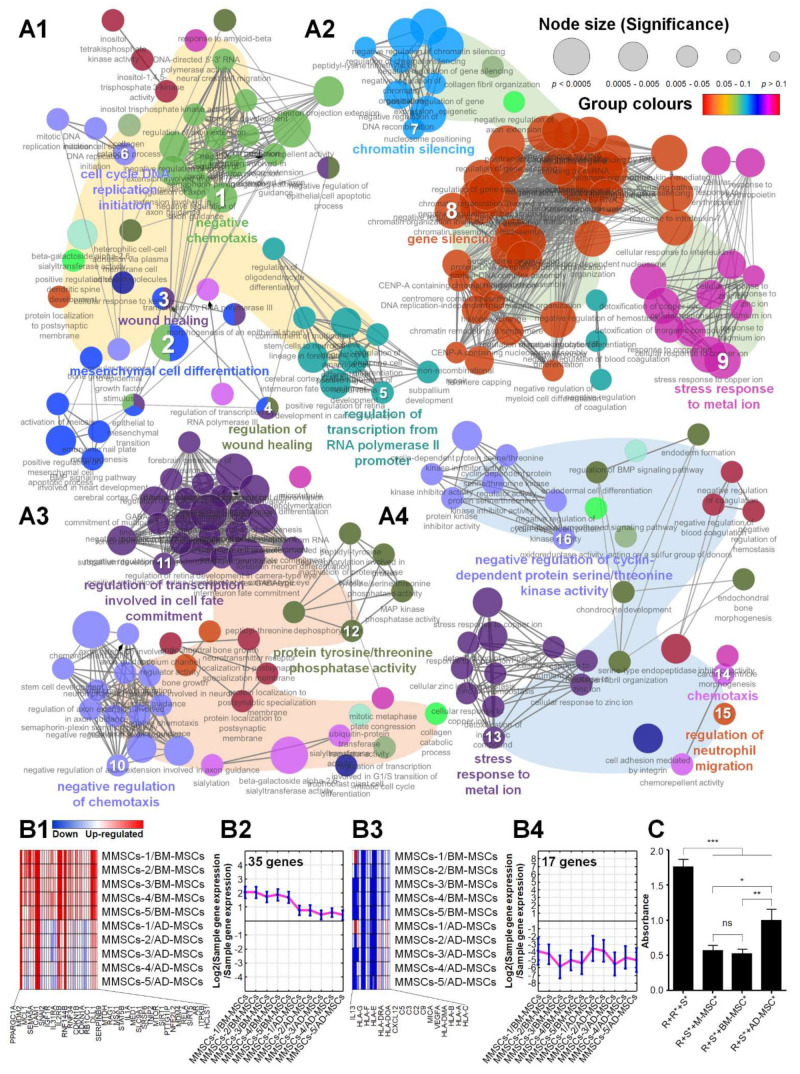
Characterization of M-MSCs compared to adult MSCs. Gene Ontology (GO) network and Enrichment Pathway analysis for M-MSCs were assessed using ClueGO. (**A1**) Upregulated differentially expressed genes (DEGs) (fold change (FC) > 4) and (**A2**) downregulated DEGs (FC < 0.25) in the M-MSCs compared to BM-MSCs were analyzed, while (**A3**) upregulated DEGs (FC > 4) and (**A4**) downregulated DEGs (FC < 0.25) in the M-MSCs compared to AD-MSCs were analyzed. Each node represents a significantly enriched biological process and shows their significance depending on their size. The gene expressions of M-MSCs were compared to adult MSCs for specific GOs. First quality threshold cluster (QTC) for GO:0043069 (negative regulation of programmed cell death) were shown as (**B1**) a heatmap and (**B2**) a line graph, while first QTC for WP2328 (Allograft Rejection (Homo sapiens)) were shown as (**B3**) a heatmap and (**B4**) a line graph. The colour scale at the top represents the relative expression levels, where red, blue, and white indicate upregulation, downregulation, and unaltered expression, respectively. (**C**) Mixed lymphocyte reaction (MLR) showed immunogenicity of M-MSCs compared to BM-MSCs and AD-MSCs (R, responder splenocytes; S, stimulator splenocytes; * mitomycin C-treated). The data are shown as mean ± SD and the significance was represented as * *p* < 0.05; ** *p* < 0.01; *** *p* < 0.001; ns, insignificant, respectively.

**Figure 2 biomedicines-10-01549-f002:**
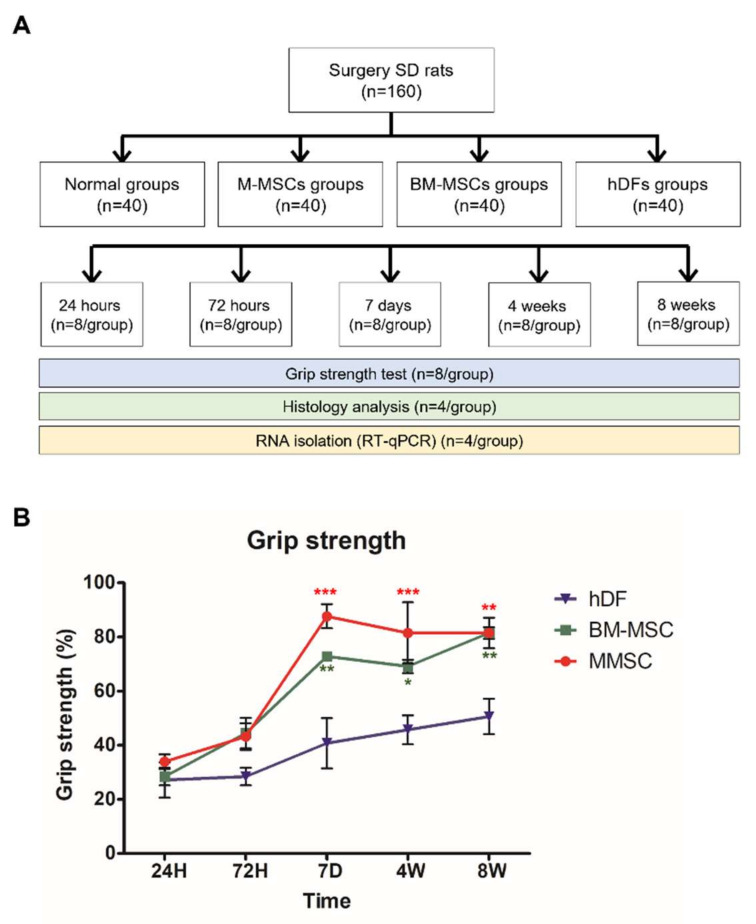
Schematic diagram of the study, effects of MSC injection on grip strength recovery after RCT induction. (**A**) Schematic diagram of the study. Seven-week-old SD rats were randomly divided into 20 groups, and cell injection was operated on the right shoulder. At 5 times, 8 mice per group were sampled and used for the experiment. (**B**) M-MSCs, BM-MSCs, or control hDFs were injected to RCT model rats. At 24 h, 72 h, 7 days, 4 weeks, and 8 weeks after RCT induction, the grip strength tests were conducted for each rat. The data are shown as mean ± SD and the significance was represented as * *p* < 0.05; ** *p* < 0.01; *** *p* < 0.001, respectively.

**Figure 3 biomedicines-10-01549-f003:**
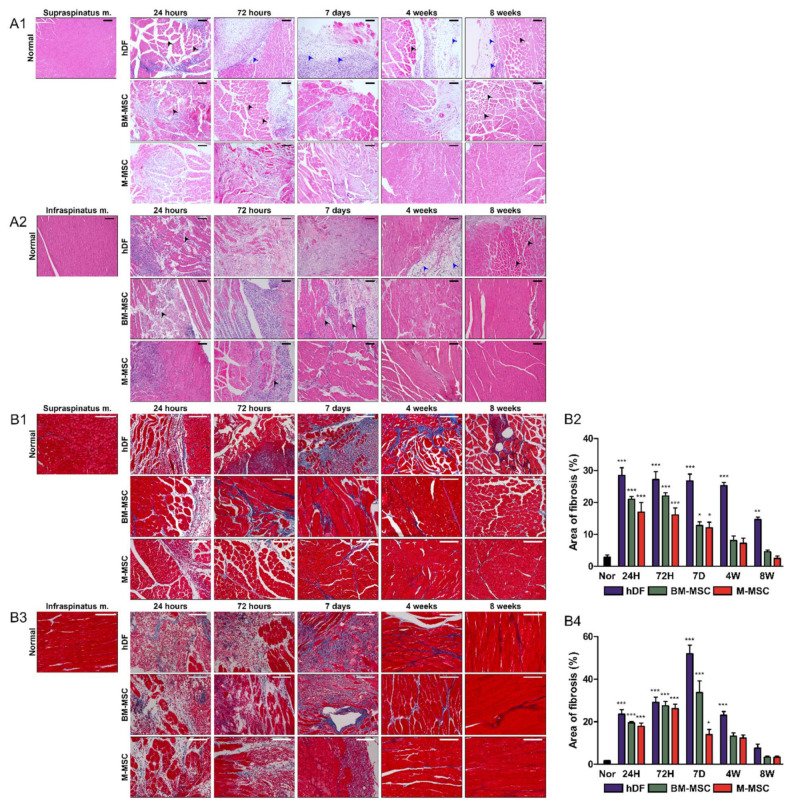
Morphological evaluation of H and E and MT-stained slides. For histopathological analysis, muscle samples were excised from hDF (control), BM-MSC and M-MSC groups at 24 h, 72 h, 7 days, 4 weeks, and 8 weeks after RCT induction. (**A1**) Supraspinatus and (**A2**) infraspinatus muscles were stained with hematoxylin and eosin. Black arrows indicate atrophy, and blue arrows indicate fat infiltrates. The scale bars are 100 μm. To quantify the fibrotic area, (**B1**,**B2**) the supraspinatus and (**B3**,**B4**) the infraspinatus muscles were stained with Masson’s trichrome. The scale bars are 100 μm. (**B2**,**B4**) The areas of fibrosis were determined as blue and were quantified in three random high-power fields (HPFs) per sample. The data are shown as mean ± SD and the significance was represented as * *p* < 0.05; ** *p* < 0.01; *** *p* < 0.001, respectively.

**Figure 4 biomedicines-10-01549-f004:**
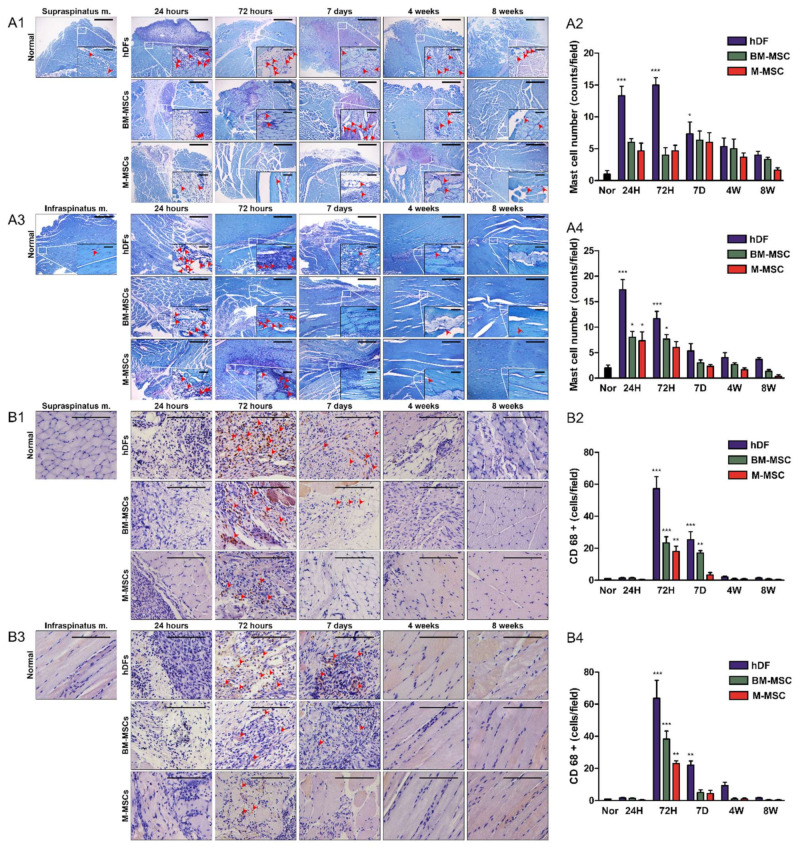
Confirmation of mast cells and CD68+ macrophages. For inflammatory cell counting, muscle samples were excised from hDF (control), BM-MSC and M-MSC groups at 24 h, 72 h, 7 days, 4 weeks, and 8 weeks after RCT induction. To confirm the mast cells, (**A1**,**A2**) supraspinatus and (**A3**,**A4**) infraspinatus muscles were stained with toluidine blue. The scale bars are 500μm (50μm for HPFs). (**A2**,**A4**) The mast cells were counted in three random HPFs per sample. To confirm the macrophages, (**B1**,**B2**) the supraspinatus and (**B3**,**B4**) the infraspinatus muscles were immunohistochemically stained with anti-CD68 antibody. The scale bars are 100μm. (**B2**,**B4**) The CD68+ macrophages were counted in three random HPFs per sample. The data are shown as mean ± SD and the significance was represented as * *p* < 0.05; ** *p* < 0.01; *** *p* < 0.001, respectively.

**Figure 5 biomedicines-10-01549-f005:**
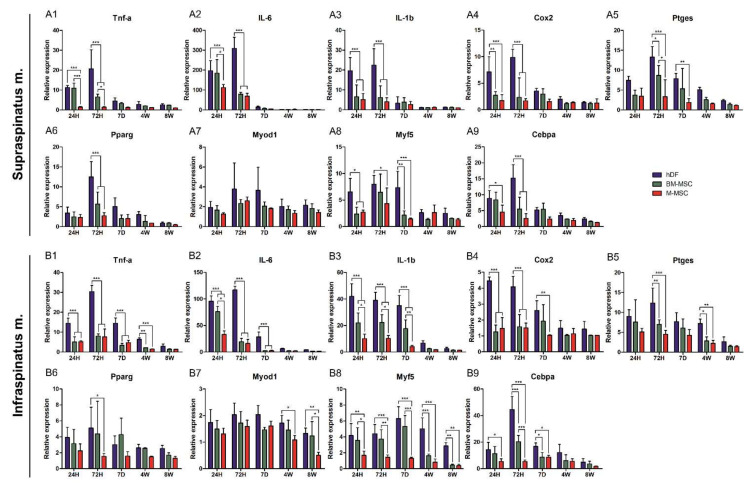
Relative expressions of genes related to inflammation, adipogenicity, and myogenicity. Supraspinatus and infraspinatus muscles were excised from hDF (control), BM-MSC and M-MSC groups at 24 h, 72 h, 7 days, 4 weeks, and 8 weeks after RCT induction. Genes related to inflammation including (**A1**,**B1**) TNF-a, (**A2**,**B2**) IL-6, (**A3**,**B3**) IL-1b, (**A4**,**B4**) Cox2 and (**A5**,**B5**) Ptges were monitored while adipogenic genes (**A6**,**B6**) Pparg and (**A7**,**B7**) Myod1, and myogenic genes (**A8**,**B8**) Myf5 and (**A9**,**B9**) Cebpa were monitored. The data are shown as mean ± SE and the significance was represented as * *p* < 0.05; ** *p* < 0.01; *** *p* < 0.001, respectively.

**Figure 6 biomedicines-10-01549-f006:**
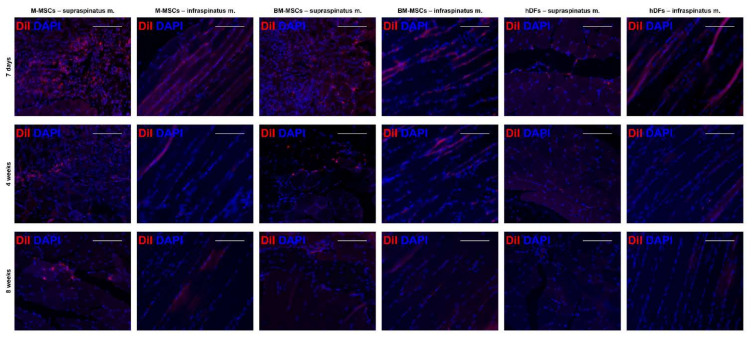
Long-term in vivo MSC tracking based on DiI. Supraspinatus and infraspinatus muscles were examined from hDF (control), BM-MSC and M-MSC groups at 7 days, 4 weeks, and 8 weeks after RCT induction. DiI+ cells were determined as red while all nuclei were shown DAPI blue. The scale bars are 100 μm.

**Table 1 biomedicines-10-01549-t001:** Primer sequences.

Gene	Forward	Reverse
GAPDH	tgccactcagaagactgtgg	ttcagctctgggatgacctt
Tnf-a	tcgtagcaaaccaccaagca	cccttgaagagaacctgggagta
IL-6	tgatggatgcttccaaactg	gagcattggaagttggggta
IL-1b	cttgtcgagaatgggcagtct	tgtgccacggttttcttatgg
Cox2	acgcctgagtttctgacaaga	taagttggtgggctgtcaatc
Ptges	gtggaagtagggtgccatgt	cagtctttggaggagccaag
Pparg	cggtttcagaagtgccttgc	ccgccaacagcttctcctt
Myod1	ccagagctgatctttgagtgg	cctgttacacccgagatctga
Myf5	aatgcaatccgctacattgag	agggcagtagatgctgtcaaa
Cebpa	ggccaagaagtcggtggata	ccaaggagctctcaggcag

## Data Availability

Not applicable.

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
