# Peer review of "Immunomodulation of Pluripotent Stem Cell-Derived Mesenchymal Stem Cells in Rotator Cuff Tears Model"

_biomedicines, 2022, doi:10.3390/biomedicines10071549_

Round 1

Reviewer 1 Report

This manuscript describes the comparison of MSCs derived from ES cells, bone marrow, and adipose for the treatments of rotator cuff tears.  Comprehensive analysis of these cells by RNA-seq revealed that ES-derived MSCs were superior to other MSCs in terms of inflammatory regulation.  The authors also showed the superiority of the ES-derived MSCs by some experimental approaches such as in vitro lymphocyte reactions, histological analyses, gene expression of transplanted tissue, and fluorescent tracking of transplanted cells.  Obtained results were persuasive and promising.  Therefore, the reviewer recommends that the authors should be accepted after the following change:

Major comments

The description of the first three paragraphs in the Discussion is quite similar with the Introduction.  The reviewer recommends that the paragraphs would be rewritten or eliminated.

In Figure 6, fluorescent microscopy of DiI-labelled cells was qualitative.  Were the authors able to analyze these images by software such as ImageJ for quantitative analyses?

Minor comments

In the section of Introduction, the authors did not explain the abbreviations “ESC-derived MSCs” and “GO”.  The abbreviations should be explained in case that the word firstly appeared.

The words in Figure A1-A4 are hard to recognize.  The authors should indicate these words clearly.

On page 14, line 449, “the concentration” should be “expression” or other words?

Author Response

Thank you for reading the manuscript carefully and for providing useful comments. Thanks to your comments, we were able to revise the manuscript to present the results more effectively. We have edited the manuscript according to your comments and have attached it as a PDF file. Thank you.

Reviewer 2 Report

Dear authors and dear Editors,

Thank you for giving me the opportunity to review this article. Here, the authors provide a concise article on using different stem cell injections to treat muscle/tendon injury - rotator cuff tears (RCT) in mice. They showed that M-MSCs reduced levels of apoptosis and exhibited significantly lower levels of immune rejection than AD-MSCs. The gene analysis was correlated with histological sections and clinical grip test. It was confirmed that the M-MSCs group showed much faster and superior muscle recovery than the BM-MSCs. M-MSCs showed inhibition for inflammation, fat infiltration, and muscle damage. The authors conclude that M-MSCs could be a potential next-generation RCT therapy. The scientific level of the article is satisfactory. There are, however, several minor issue s that must be addressed. These are the following.

Sincerely.

Please unify reference style. I suppose the references should be numbered. But you also use the name-date style, e.g., Davies et al. 2015, which I think is the wrong style – Line 46.

Please explain the abbreviation at its first appearance in the text: ESC - Line 61

Please explain the abbreviation at its first appearance in the text: GO analysis - Line 77.

Please explain the abbreviation at its first appearance in the text: BM-MSCs - Line 78.

Wrong capital: and The ability – and the ability - Line 78

Please explain the abbreviation at its first appearance in the text: AD-MSCs - Line 109

Missing space: function cart.Correlation - Line 134

Missing upper case?: 105 cells/100 µL/well - 105 cells/100 µL/well - Line 170

I am not sure about this sentence on Line 182-183: One hundred twenty rats were randomly divided into 20 groups (M-MSC, BM-MSC, Control, Normal for each time; n = 8). I think that you meant 160 rats? What about this: One hundred and sixty rats were randomly divided into 4 groupsets (M-MSC, BM-MSC, Control, Normal) and 5 intervals, thus creating 20 groups (for each time n = 8).

Please explain what was done in these two groups: Control, Normal. Was that a rotator cuff tear without cell treatment? Or a sham operation (incision without a tear)? Or placebo/saline injection? It is not clear from the text but essential for further reading and understanding  – Line 183

Wrong punctuation: the greater tuberosity and. After surgery - Line 188

Line 189: spelling and spacing error: and hDFswere injected

Please explain the abbreviation at its first appearance in the text: hDF. Line 189

What is the hDF treatment? Is that some control/placebo treatment? This is essential for further reading. Line 189

Line 204: Grip strength Measurement in rats – please explain how did you ensure that the rat grabbed the mesh with the operated limb? Was that the right front limb?

Line 221: Punctuation error: Deajeon, Korea). According

Line 263: Spelling error: masson’s trchrome – Masson´s trichrome

Line 375: the sign “µ” has a different font. The same further. Please ensure that the font is the same throughout the whole text.

Line 381 Spelling error: by day 7 days

Line 392: In the subscapularis muscle… Do you mean infraspinatus muscle? I assume that subscapularis and infraspinatus are different muscles.  In the previous and following text, you deal with supraspinatus and infraspinatus muscles only. Please make it clear.

Line 400: Please explain abbreviations at their first appearance in the text: TB staining and IHC

Line 402: less than 5 mast cells were found… Do you mean less than 5 cells per area? Per visual filed? Please explain.

Line 404: and was slightly different compared with the normal group… There is no subject. What was slightly different? And was it slightly higher or lower compared with the normal group? Please make this clear.

Line 472: more than a third or a tenth of all patients… I do not understand. Is that more than a third or is than a tenth? Please explain.

Line 477: Please explain the abbreviation: NSAIDs

Line 489: MMSC Please use consistently the same form, such as M-MSC. The same is further in the text.

Line 571: Please explain the abbreviation: iPSCs

Author Response

(The authors gave the same response as above.)

Reviewer 3 Report

In this manuscript, titled “Immunomodulation of pluripotent stem cell-derived mesenchymal stem cells in rotator cuff tears model”, the authors assess the ability of mesenchymal stem cells of various lineages (ESC-derived versus BM-derived) to control inflammation and restore rotator cuff after injury. While interesting, the authors never fully confirmed whether the M-MSCs were truly multipotent. There were a variety of issues with the animal model, and lack of justification for several of the components within the study (i.e. the lack of an injury-only control, why include hDFs, why inject into the muscle when the injury was in the tendon, etc.). Based on the data presented in the Figures, there does not appear to be much of a difference in the response of the injury to BM-MSCs and M-MSCs. In this reviewer’s opinion, the significant flaws present in the manuscript greatly diminish its impact.

Major Concerns:

  1. Part of the introduction is confusing. The comparison between “hPSCs”, MSC, and then “ESCs” (as if they are separate from hPSCs) is a bit jarring and doesn’t make intuitive sense as to why these would all be compared together.
  2. It appears there are 3 groups mentioned on lines 182-183, however using both “Control” and “Normal” separately implies there are 4 groups (M-MSC, BM-MSC, Control, Normal for each time). The authors should correct this to avoid confusion. Further, these do not describe the use of hDFs.
  3. Why inject hDFs into the wound site, and what is their source?
  4. Why is there no injury-only group, to evaluate the improvement with respect to the injury and not just healthy? This is a significant concern and including this group is necessary to fully interpret the treatment groups.
  5. What is the rationale for injecting the cell therapy into the belly of the muscle, rather than by the tendon (which, presumably, is what was injured)?
  6. The rotator cuff tear surgery is difficult to understand. What was “partially detached”, and how did you ensure repeatability of these measurements?
  7. How did the authors control for a rejection response from the xenogeneic cell source? Sprague Dawley rats are immune-competent and injecting human cells would likely cause a rejection response. If the authors used a drug to mitigate this, would this impact their inflammation results?
  8. Why use a grip strength measurement to study rotator cuff recovery? Grip requires fore-limb muscles and is not dependent on the rotator cuff.
  9. Did the authors assess multi-potency of their M-MSCs? These data should be included in the manuscript.
  10. The authors claim that M-MSCs showed reduced levels of apoptosis compared to adult MSCs (lines 334-335) – how was this determined? These data need to be included in the manuscript as well.
  11. Did the authors normalize their grip strength data to wild type/healthy animals? It is not clear what “normal” means (Section 3.2). Additionally, quicker recovery of grip strength does not necessarily mean that one group had “superior muscle recovery” (line 352), such an assessment requires information regarding the muscle morphology.
  12. It is not clear why muscle is being evaluated morphologically if the surgical intervention was done in the tendon.
  13. How is muscle atrophy being determined? It is more standard to measure cross-sectional area of the myofibers. Based on the images provided in Fig 3, there is little to no fatty infiltration observable (although there are a lot of mononuclear cells present), and the muscle appears to be alternatively sectioned in cross-section and longitudinally, making it difficult to assess morphology. More thorough characterization and inclusion of methods is necessary.
  14. Based on the data presented, the claim that M-MSCs showed a less inflammatory response than BM-MSCs seems slightly overstated.
  15. Is there evidence that ESCs/M-MSCs will be immunoprivileged if implanted? A major advantage of BM-MSCs or AD-MSCs as a treatment is that they can be extracted from the patient, cultivated in vitro, and then implanted.
  16. How successful is the differentiation protocol to develop M-MSCs? A major comment by the authors is that M-MSCs consistently express the same genes, which is not surprising given that they were harvested from a single cell line. What happens if M-MSCs are derived from a different cell population, would they be equally as consistent?

Minor Concerns

  1. Mesenchymal stem cells are multipotent, they are not pluripotent (line 53).
  2. Why two separate citation methods? The sources cited in parentheses are not included in the references section.
  3. The small text in Fig 1 is somewhat illegible and needs to be enlarged.
  4. The CD68 images (Fig 4 B1, B3) should be reproduced with higher brightness.
  5. There are numerous typos and confusing word selections throughout the manuscript. A more thorough proof reading is required.

Author Response

(The authors gave the same response as above.)

Reviewer 4 Report

The authors investigated the use of M-MSCs as stem-cell source for RCT treatment. In the study they compare the transcriptomic profile of M-MSCs with BMSC and ASC. Moreover they assessed their immunomodulatory effect by using a RCT in vivo model.  

Please find below some comments/suggestions which might help to improve the quality of the manuscript.

- lines 53 ; please check the MSCs definition. They are a multipotent stem cells;

- lines 46,48,49, 57 please check the reference format

- Please provide the explanation for each abbreviation when it compare in the text (i.e line 61, ESC-derived MSCs)

-lines 60-64 please clarify

- In the introduction section the aim, objectives, and methods of the study are missing. Moreover, the authors should not describe your results in this section (76-85 lines). Please complete this section.

- lines 77-78 typing error

- In the statistical analyses section the statistical test used must be add.

- the images in Figure 1 ( above all B4 and C) and in Figure 4 are not visible enough. Please increase the dimension for example by split them in more figures.

-lines 380 typing error

Author Response

(The authors gave the same response as above.)

Round 2

Reviewer 3 Report

In this revised manuscript, the authors have made some revisions to the document. The context they provided in the response to reviewers was thorough, however, not much of that content was reproduced in the manuscript itself. Inclusion of those discussion points, and in particular the ones listed below, is necessary:

1.      If sham (injury only, negative control) data is available, this would greatly strengthen the conclusions of the authors. Comparisons to healthy or fibroblast injections, which is not a standard clinical practice, is not as

2.      The annotated Fig 3 is not included in the revised manuscript.

3.      Include the rationale for hDFs as a treatment group and their injection site.

4.      Include discussion of M-MSCs functioning as a paracrine mechanism.

5.      A reference to the previous publication assessing the multi-potency of the M-MSCs should be included in the manuscript.

6.      Include a discussion regarding heterogeneity (or similarity) of M-MSCs harvested from different sources, as this is a major comment made by the authors and should be addressed within the discussion.

7.      While the authors explained that the M-MSCs are putatively immunoprivileged, was there an immune rejection response (independent of the inflammation reported in the manuscript) to the hDFs?

Author Response

Thank you for your kind comments and quick review. The manuscript has been revised and added based on your comments. You can find the details below. Thank you

  1. If sham (injury only, negative control) data is available, this would greatly strengthen the conclusions of the authors. Comparisons to healthy or fibroblast injections, which is not a standard clinical practice, is not as

-> According to your opinion, H&E data and q-PCR data for the sham group have been added as a supplementary figure. For the sham group, only H&E figure and q-PCR data for inflammation-related factors are prepared. You can check the added data in the supplementary file figure 3. Thank you.

  1. The annotated Fig 3 is not included in the revised manuscript.

-> Thanks for your comments. The arrow of the figure provided in the last revision has been modified to match the arrow style of other figures. The legend of the changed figure has also been modified. The figure and legend can be found in the manuscript. (p.12, in the Result section) Thank you.

  1. Include the rationale for hDFs as a treatment group and their injection site.

-> Thanks for the nice comments. According to your opinion, the reason for using hDFs and the contents of the injection site were additionally written in the manuscript. (highlighted with blue color in p.17, Line 502, in the Discussion section) Thank you.

  1. Include discussion of M-MSCs functioning as a paracrine mechanism.

-> Thanks for the helpful comments. Based on your comments, we have added the content to the manuscript. (highlighted with blue color in p.17, Line 502, in the Discussion section) Thank you.

  1. A reference to the previous publication assessing the multi potency of the M-MSCs should be included in the manuscript.

-> Thanks for the nice comments. Based on your comments, we have included in the manuscript a reference to a previous publication assessing the multi-potency of M-MSCs. (highlighted with blue color in p.17, Line 492, in the Discussion section) Thank you.

  1. Include a discussion regarding heterogeneity (or similarity) of M MSCs harvested from different sources, as this is a major comment made by the authors and should be addressed within the discussion.

-> Thanks for the helpful comments. We wrote the answer and reference of the content in the manuscript. (highlighted with blue color in p.17, Line 497, in the Discussion section) Thank you.

  1. While the authors explained that the M-MSCs are putatively immunoprivileged, was there an immune rejection response (independent of the inflammation reported in the manuscript) to the hDFs?

-> We totally agree with the importance of immune rejection in cell therapy and understand your question about this. Of course, it's important to identify different mechanisms for immunization rejection, but we do not think this is entirely consistent with the purpose of our study.

The research on this is not an additional part of our present study, but we would like to conduct a separate mechanism study on the immune rejection of cell therapy. I ask for your understanding.

Thank you.
